# Fecal Levels of Lactic, Succinic and Short-Chain Fatty Acids in Patients with Ulcerative Colitis and Crohn Disease: A Pilot Study

**DOI:** 10.3390/jcm10204701

**Published:** 2021-10-13

**Authors:** Olga Kaczmarczyk, Agnieszka Dąbek-Drobny, Michał Woźniakiewicz, Paweł Paśko, Justyna Dobrowolska-Iwanek, Aneta Woźniakiewicz, Agnieszka Piątek-Guziewicz, Paweł Zagrodzki, Tomasz Mach, Małgorzata Zwolińska-Wcisło

**Affiliations:** 1Department of Gastroenterology and Hepatology, Medical College, Jagiellonian University, 30-060 Krakow, Poland; okaczmarczyk@su.krakow.pl (O.K.); agnieszka.guziewicz@onet.eu (A.P.-G.); tmach@su.krakow.pl (T.M.); 2Unit of Clinical Dietetics, Department of Gastroenterology and Hepatology, Medical College, Jagiellonian University, 30-060 Krakow, Poland; a.dabek-drobny@uj.edu.pl; 3Department of Analytical Chemistry, Faculty of Chemistry, Jagiellonian University, 30-060 Krakow, Poland; michal.wozniakiewicz@uj.edu.pl (M.W.); aneta.wozniakiewicz@uj.edu.pl (A.W.); 4Department of Food Chemistry and Nutrition, Medical College, Jagiellonian University, 30-060 Krakow, Poland; paskopaw@poczta.fm (P.P.); justyna.dobrowolska-iwanek@uj.edu.pl (J.D.-I.); pawel.zagrodzki@uj.edu.pl (P.Z.)

**Keywords:** Crohn’s disease, ulcerative colitis, inflammatory bowel disease, short-chain fatty acids, succinic acid, lactic acid

## Abstract

Intestinal dysbiosis plays a crucial role in the development of inflammatory bowel disease (IBD), including Crohn’s disease (CD) and ulcerative colitis (UC). The importance of bacterial metabolites, such as short-chain fatty acids (SCFAs), succinic and lactic acids, as well as environmental factors that affect their production in the course of IBD, remains unclear. The aim of this study was to evaluate a profile of organic acids in the stool of CD and UC patients with different disease activity. We assessed a correlation between used medications, patient’s diet, and SCFA levels. A total of 35 adult patients were included in the study. We did not observe significant differences in the levels of organic acids between the CD and UC groups, irrespective of disease activity, and a control group. However, propionic acid levels were higher in IBD patients who received trimebutine vs. those who did not (*p* = 0.031). Higher isobutyric acid levels were observed in patients treated with biological drugs compared with those without such treatment (*p* = 0.014). No significant correlations were found between organic acid levels and the frequency of dietary fiber consumption. Our results reveal a new link between medication use and organic acid levels in patients with IBD.

## 1. Introduction

Inflammatory bowel disease (IBD), including Crohn’s disease (CD) and ulcerative colitis (UC), is a noninfectious chronic inflammation of the gastrointestinal tract. It is characterized by periods of exacerbation and remission. The most common clinical symptoms in patients with CD are abdominal pain and anemia, or for those with UC, bloody diarrhea. The frequency of IBD has been increasing worldwide. According to a recent study, the global number of people with IBD has risen from 3.7 million to over 6.8 million over the past 30 years, which translates to an increase of 85.1%. The global total number of deaths has increased by 67.0% [1].

The etiopathogenesis of IBD is complex and remains unclear. The mutual influence of genetic and environmental factors as well as microbial imbalance (dysbiosis) leads to immune dysregulation and disease development [2,3]. Based on the most recent research, it seems that environmental factors that affect the composition of the intestinal microbiota play a major role in the growing incidence of IBD and affect the course of the disease [2,4]. As a consequence, the production of beneficial bacterial metabolites, in particular short-chain fatty acids (SCFAs), is disrupted.

Short-chain fatty acids are the main bacterial products formed by the fermentation of indigestible dietary fiber in the intestine [5]. They exert numerous positive effects as the primary energy source for colonocytes. They also have anticancer properties, induce mucus production, regulate intestinal homeostasis by activating the inflammasome and producing interleukin IL-18, reduce intestinal inflammation, and improve gut barrier integrity [6,7]. The main substrates for bacterial fermentation and SCFA production are resistant starch, oat bran, inulin, wheat, cellulose, guar gum, and pectin [7]. In healthy individuals, the most abundant bacterial phyla in the intestines that produce SCFAs are Bacteroidetes and Firmicutes [7]. In the lumen of the gastrointestinal tract, SCFAs occur as anions, mainly acetate, propionate, and butyrate, which account for over 95% of all acids [5]. According to previous studies, butyric acid has a key role in the maintenance of gut homeostasis [8]. By binding G protein-coupled receptors and inhibiting histone deacetylase, SCFAs exert anti-inflammatory activity that appears to be relevant in protecting against the development and exacerbation of inflammatory disorders [9]. Additionally, other organic acids produced in the intestine by bacteria, such as lactic and succinic acids, are important SCFA precursors that may play a relevant role in health and disease. Succinic acid has attracted considerable attention as a proinflammatory mediator in intestinal inflammation, and as a profibrotic marker. Recent research indicated that another SCFA precursor, lactic acid, may be a signaling molecule that influences the immune regulation of the intestinal mucosa [8].

In recent years, there has been a growing interest in the beneficial role of SCFAs in the course of dysbiosis-related diseases, in particular IBD. A previous study showed that IBD is associated with a depletion of SCFA-producing microorganisms, such as *Faecalibacterium prausnitzii* and *Roseburia hominis*, and a reduction in SCFAs levels [10]. Moreover, the composition of beneficial SCFAs may be affected by modifiable environmental factors, such as pharmacologic interventions, dietary habits, smoking, and level of stress exposure. Dietary modifications commonly used in IBD are based on the amount of fiber in diet (e.g., a low-residue diet and an easily digestible diet) and may affect the production of SCFAs [11]. The effect of diet on SCFAs production in IBD patients seems to be relevant; therefore, the exact link needs to be confirmed.

Medications used to treat IBD include 5-aminosalicylic acids (5-ASA), corticosteroids, as well as immunosuppressive or biological drugs. However, their effect on gut microbiome and SCFA profile in IBD remains unknown.

Despite previous evidence that patients with IBD have altered fecal SCFAs levels, the complete SCFAs profile and the related factors are not fully understood. The quantitative and qualitative changes that occur in IBD patients as compared with healthy individuals have not been determined. It seems that the changes in SCFAs levels may be related to the type of IBD, disease activity, and various environmental factors. Moreover, the role of some organic acids, such as succinic and lactic acids, in intestinal inflammation is unclear.

The aim of our study was to investigate the quantitative and qualitative changes in the levels of SCFAs, branched-chain fatty acids (BCFAs) such as isovaleric and isobutyric acids, as well as SCFAs precursors (succinic and lactic acids) in the stool of patients with different activity of UC and CD, both in remission and active disease. Moreover, we assessed correlations between fecal organic acid levels and medical therapy (biological, 5-ASA, and immunosuppressive drugs) as well as diet. Finally, we aimed to determine correlations between fecal organic acid levels and laboratory parameters such as complete blood count, serum C-reactive protein (CRP) levels, and fecal calprotectin levels.

## 2. Materials and Methods

### 2.1. Study Population

The diagnosis of UC and CD was confirmed by clinical evaluation as well as endoscopic, histological, radiological, and biochemical studies. Disease activity was determined according to Mayo criteria for UC patients and the Crohn’s Disease Activity Index (CDAI) for CD patients. The control group included patients with functional bowel disorders, without organic and inflammatory lesions in the large intestine on colonoscopy, and who did not meet Rome IV diagnostic criteria for irritable bowel syndrome. Exclusion criteria were as follows: acute viral and bacterial infections of the gastrointestinal tract, pregnancy, malignancy, diabetes mellitus, obesity (body mass index ≥ 30 kg/m^2^), cardiovascular disease, severe chronic liver disease and chronic renal failure, tobacco smoking, alcohol abuse, intake of probiotics, synbiotics, and dietary supplements containing sodium butyrate, as well as total or partial parenteral nutrition.

In all participants, clinical information was obtained, followed by colonoscopy or sigmoidoscopy to assess inflammatory lesions in the intestine. Blood samples were obtained to measure complete blood count and CRP levels. Stool samples were collected to identify selected organic acids, including butyric, propionic, acetic, valeric, isovaleric, isobutyric, succinic, and lactic acids, as well as calprotectin levels. Biochemical studies, including the measurement of complete blood count, serum CRP levels, and fecal calprotectin levels were performed at the Department of Diagnostics of the University Hospital in Krakow, Poland, in accordance with relevant laboratory procedures.

Moreover, participants were asked to complete a nutritional questionnaire, designed by ourselves in consultation with a clinical dietitian. The questionnaire assessed the frequency of dietary fiber consumption and presence of dairy products in diet over the past month. The questions concerned the frequency of consumption of raw fruit and vegetables, dried fruit, wholegrain products, and dairy products. Patients were divided into three groups depending on the frequency of dietary fiber consumption: high-fiber diet (high-fiber products consumed everyday), regular-fiber diet (high-fiber products consumed several times a week), and low-fiber diet (high-fiber products consumed several times a month or less). With regard to dairy products, patients were divided into two groups depending on whether they consumed dairy products or not.

Written informed consent was obtained from all participants before enrollment to the study. The study was approved by the Bioethics Committee at Jagiellonian University in Krakow, Poland (no. 1072.6120.18.2018), and was performed in accordance with the Declaration of Helsinki.

### 2.2. Measurement of Organic Acids in Stool Samples

Fecal samples were collected intro sterilized plastic cups with screw caps and stored at a temperature of −80 °C until analysis. Samples were prepared and then extracted at the Department of Food Chemistry and Nutrition, Faculty of Pharmacy, Jagiellonian University Medical College, Krakow, Poland. The preparation of each stool sample included drying, milling, and subsequent extractions. The extraction conditions were optimized by selecting the shaking time, ultrasound exposure time of the sample, and the number of extractions according to a previous study [12]. All obtained extracts were centrifuged and stored at a temperature of −20 °C until further analysis.

The short-chain fatty acids pilot investigation in stool extracts were made using an isotachophoresis system (Electrophoretic Analyser EA 202 M, Villa Labeco, Spisska Nova Ves, Slovakia) with a conductivity detector.

Separation and determination of selected organic acids (succinic, acetic, lactic, propionic, butyric, isobutyric, valeric, and isovaleric acids) were performed using capillary electrophoresis, PA 800 plus Pharmaceutical Analysis System (Beckman Coulter, Brea, CA, USA), furnished with an ultraviolet spectrophotometric detector at the Laboratory for Forensic Chemistry, Faculty of Chemistry, Jagiellonian University. All obtained extracts were filtered through modified cellulose filters (0.45 µm, 25 mm) and then analyzed by capillary electrophoresis with spectrophotometric UV detection. The separation process was conducted in a fused-silica capillary (internal diameter, 75 µm; total length, 60 cm) with a high voltage of −30 kV (reverse polarity mode, cathode at inlet) applied at 25 °C and a background electrolyte composed of 1% methyl-β-cyclodextrin in a commercially available buffer (Anion Kit 5, Analis, Namur, Belgium). Samples were injected hydrodynamically (0.5 psi at 34 mbar for 8 s). The spectrophotometric detection was performed at 230 nm using an indirect detection mode. Fecal SCFAs levels were expressed as mean μg per gram dry of weight feces. The limit of quantification (LOQ) was 26 µg/g (taken as the lowest concentration of the standard solution from the calibration curves measured for a given analyte—7.8 µg/mL, taking into account the mass of the sample 300 mg), and precision, calculated as the RSD of repeated measurements, (*n* = 9) did not exceed 15%.

### 2.3. Statistical Analysis

Descriptive statistics were calculated for all organic acids in the whole study group. To calculate the mean values, data were logarithmically transformed and retransformed after calculations, as the parameters had non-Gaussian distribution. The differences between groups were assessed using Student’s-*t* test, Welch test, Mann–Whitney test, or Kruskal–Wallis with a post-hoc Dunn test. The Levene test was used to assess the equality of variances in study groups. A probability level of *p* < 0.05 was considered to be significant. The principal component analysis (PCA) model was used to describe the correlation structure between parameters in the whole study group. The parameters with large loadings on the first two principal components (>0.3) were assumed to be correlated with one another. To express the strength of bivariate associations, for the pairs of correlated parameters the algebraic products of their corresponding loadings and the cosine of the corresponding angle were calculated (these coefficients are called the correlation weights). The “corresponding angle” means the angle determined by the two lines connecting the origin with coordinates of both parameters on the PCA loadings plot. The PCA approach was also applied to check whether the clusters of patients with UC or CD with varying severity, as well as controls, appear in the PCA score plot.

Statistical analyses were carried out using STATISTICA v.12 (StatSoft, Tulsa, OK, USA), Graph Pad Prism v.3.02 (GraphPad Software, San Diego, CA, USA), and SIMCA-P v.9 (Umetrics, Umeå, Sweden). The software delivered by MP System Co. (Chrzanów, Poland) was used to calculate correlation weights for the pairs of parameters in the PCA model.

## 3. Results

### 3.1. Study Population

The study included 35 patients hospitalized at the Department of Gastroenterology and Hepatology of the University Hospital in Krakow, Poland. There were 23 patients with UC (13 men and 10 women; median age, 32 years), 8 patients with CD (5 men and 3 women; median age, 33.5 years), and 4 controls (2 men and 2 women; median age, 48 years). During the study, IBD patients continued their current treatment with tumor necrosis factor (TNF) inhibitors (infliximab or adalimumab), 5-ASA, corticosteroids, and immunosuppressants (azathioprine or 6-mercaptopurine). Because a significant proportion of participants used trimebutine, these patients constituted a separate subgroup. The demographic characteristics of the study groups are presented in Table 1.

### 3.2. Fecal Organic Acid Levels in IBD Patients and the Control Group

All studied organic acids (acetic, lactic, propionic, succinic, butyric, isovaleric, isobutyric, and valeric) were identified in stool samples. The median levels of organic acids in the study groups are presented in Table 2. In patients with mild-to-moderate UC and inactive CD, as well as in controls, the highest median concentration was observed for acetic acid, while patients with severe UC and active CD had the highest median levels of lactic acid. No differences were observed in the concentrations of organic acids between the CD and UC groups, irrespective of disease activity, and the control group. However, the butyric acid level was lower in patients with active CD compared with those with inactive CD (Table 2), but the difference was not significant. The same trend was observed for patients with severe UC compared with those with moderate UC (Table 2). Interestingly, the concentration of succinic acid increased in patients with active CD compared with those with inactive CD and controls (Table 2), although the differences were not significant.

### 3.3. Fecal Organic Acid Levels and Medications

Propionic acid levels were higher in IBD patients who received trimebutine compared with those who did not take it (median 440.8 µg/g vs. 222.9 µg/g; *p* = 0.031). In addition, higher levels of isobutyric acid were observed in patients treated with biological drugs compared with those who did not receive such treatment (median 49.1 µg/g vs. 32.2 µg/g; *p* = 0.014). No differences were observed for organic acid levels in relation to corticosteroid, antibiotic, and immunosuppressive treatment.

### 3.4. Fecal Organic Acid Levels and Dietary Factors

No significant associations were found for any of the studied organic acids and the frequency of dietary fiber consumption. Similarly, no differences were observed between SCFA levels in relation to dairy product consumption.

### 3.5. Fecal Organic Acid Levels and Calprotectin

Median lactic acid levels were higher in IBD patients with high calprotectin levels (>150 µg/g) than in those with moderate calprotectin levels (50–150 µg/g): 997.1 µg/g vs. 28.5 µg/g (*p* <0.01). Interestingly, there were no significant differences in the levels of other organic acids in relation to calprotectin levels.

### 3.6. PCA Model

The PCA model had two significant components with an eigenvalue of 3.93 and 2.53. This model explained 53.8% of the original variance. The loadings for the first two principal components are shown in Figure 1. The first principal component in this model was loaded positively by hemoglobin, hematocrit, and red blood cells (RBCs), which were correlated with one another. Thus, the highest positive correlation weights based on this component were revealed for these three parameters. The first principal component had negative weights for platelets and CRP, suggesting negative correlations between these two parameters and hemoglobin, hematocrit, and RBCs. The second principal component was loaded positively again by hemoglobin, hematocrit, and RBCs, and, additionally, by lactic acid, while negatively, mainly by age (Figure 1, Table 3). Visual analysis of the score scatterplot of the PCA model did not reveal any separate and homogenous clusters of subjects (Figure 2). In particular, the PCA model showed, in the context of the 12 most statistically relevant parameters, that the concentrations of most SCFA did not differentiate patients with different diagnoses. The only parameter in this group (lactic acid concentration) was more strongly correlated with other parameters and was also useful for discriminatory purposes.

## 4. Discussion

In this study, we comprehensively evaluated changes in fecal organic acid levels in the course of disease in patients with CD, UC, and the control group. In addition, we examined the association between fecal organic acid levels and some environmental factors that may affect their production, including dietary factors and different medications used in IBD. Our results revealed a novel link between fecal levels of organic acids and medications used in patients with IBD. For the first time, we demonstrated an association between isobutyric acid levels and biological therapy as well as between propionic acid levels and trimebutine use. In addition, we showed that increased lactic acid levels were associated with an exacerbation of IBD as determined by biochemical studies and clinical assessment.

Our study showed that butyric acid levels did not differ between patients with active and inactive UC and CD and the control group. However, we observed a trend for lower butyric acid concentrations in patients with active CD compared with those with inactive disease. The same was observed for patients with severe UC compared with those with moderate disease. This phenomenon can be explained by the fact that the deficiency of certain beneficial microbes such as Faecalibacterium prausnitzii in IBD patients, especially those with severe course of disease, leads to a reduction in butyrate levels and contributes to the development and maintenance of inflammation in IBD. Butyric acid has anti-inflammatory, antioxidant, and regenerative properties, thus it seems that its reduced level may contribute to exacerbation of the disease. However, we did not observe significant differences in the levels of SCFAs and BCFAs between patients with UC and CD with different disease severity. Previous studies demonstrated reduced SCFA levels in patients with IBD and a correlation between reduced levels and disease activity [13,14]. Nevertheless, the results so far are inconclusive. The only meta-analysis investigating the SCFA profile in IBD patients reported that the butyrate level was lower in CD patients, although an analysis accounting for different disease severity could not be performed due to limited data available [14]. A similar trend for lower butyrate levels was also reported in patients with severe UC [14]. These results are in line with our findings. However, in contrast to our data, previous studies demonstrated reduced acetate, valerate, and propionate levels both in UC and CD patients [14,15,16]. These discrepancies indicate the need for studies on a larger number of patients divided according to the type of IBD and disease severity to elucidate abnormalities in the SCFA profile in this population. It is especially important to identify the subgroups of IBD patients with SCFA deficiency and to understand the exact mechanism of SCFA action in order to select patients for SCFA supplementation.

Our findings showed that there is no significant difference in the concentration of succinic acid in patients with UC and CD with different disease severity. However, succinic acid levels were higher in all UC and CD subgroups as compared with the control group. Previous studies also noted increased succinic acid levels in IBD patients [16,17]. It is likely that excessive succinic acid levels due to the activation of the *SUCNR1* gene expressed in several immune cells exert significant proinflammatory and profibrotic effects in IBD [18]. The *SUCNR1* activation induces proinflammatory signaling pathways, such as dendritic cell activation and migration to lymph nodes as well as production of proinflammatory cytokines [18]. However, its mechanism of action is still poorly characterized.

We also observed a trend for higher lactic acid levels in patients with active UC and CD compared with those with inactive disease and controls. Interestingly, in patients with severe UC and active CD, lactic acid reached the highest levels of all organic acids studied. Moreover, our data demonstrated a positive correlation between fecal levels of calprotectin, a marker of intestinal inflammation, and lactic acid in IBD patients. In line with our findings, previous research demonstrated higher fecal lactate levels both in active UC and CD [19,20]. Presumably, higher lactic acid levels are associated with diarrhea and more severe inflammation in autoimmune disorders [20,21]. Our findings indicate a link between lactic acid levels and the severity of IBD. It is possible that dysbiosis, commonly observed in active IBD, may prevent conversion of lactate into butyrate, leading to an increase in lactate acid levels and a decrease in butyrate acid levels. However, we noted a positive correlation between lactic acid levels and complete blood count parameters such as hemoglobin, hematocrit, and RBC count. However, changes in red cell parameters can be due to numerous factors, such as disease duration, treatment, and complications of IBD, apart from disease activity alone. It is likely that high lactate levels observed during exacerbation of CD and UC induce intestinal inflammation as a key proinflammatory factor [21,22]. It appears that understanding the role and mechanisms behind the accumulation of succinic and lactic acids in patients with IBD may provide opportunities for developing a new treatment strategy.

It is well known that medications may significantly alter the composition of gut microbiota; however, their exact effects on gut microbes and the production of microbial metabolites such as SCFAs are not well understood [23]. To our knowledge, we are the first to report increased levels of isobutyric acid in IBD patients treated with biological drugs compared with patients who did not receive such treatment. We did not observe differences for the other organic acids. These findings are relevant in the context of the most recent reports that highlighted a possible association between SCFAs and the efficacy of IBD therapy [24]. Aden et al. [24] noted that the levels of SCFAs, particularly butyrate acid and substrates involved in butyrate synthesis, were significantly associated with clinical remission after treatment with TNF inhibitors and may be a clinical marker of therapeutic efficacy in IBD. In addition, in patients who achieved remission after biological therapy, a relevant increase in the fecal levels of butyrate acid was observed [24]. Moreover, Aden et al. [24] reported that TNF-inhibitor therapy restored the balance of gut microbiome, especially SCFA-producing bacteria. All this evidence suggests that patients after TNF-inhibitor therapy should have higher levels of SCFAs. We showed that among fecal organic acids, only isobutyric acid levels were significantly increased, which may be due to a small sample size as well as differences in the duration and type of biological therapy. Another possible explanation is the different types of diet in this group of patients. Additionally, limited carbohydrate intake may cause increased production of isobutyric acid [25].

To the best of our knowledge, no previous studies have investigated the association between trimebutine use and SCFA levels in patients with IBD. Therefore, our study is the first to report that propionic acid levels were significantly higher in IBD patients who received trimebutine compared with those who did not. Trimebutine alters intestinal transit time, which can disrupt the production and absorption of SCFAs [26]. It is possible that patients taking trimebutine have a shorter colon transit time and lower absorption, and, consequently, higher SCFA levels in stool. However, we expected increased levels of other SCFAs.

In our study, the frequency of dietary fiber consumption from wholegrains, raw vegetables, and dried and fresh fruit, as well as dairy product consumption showed no correlations with the SCFA profile or fecal calprotectin levels. In contrast to our results, recent studies demonstrated that increased dietary fiber intake is associated with higher SCFA levels in IBD patients [27,28]. Moreover, previous research showed that an intervention with an oat bran-enriched diet increased butyrate acid levels in the stool of patients with UC [29]. In addition, oligofructose supplementation reduced dyspeptic symptoms and was related to lower fecal calprotectin levels in acute UC [30]. This discrepancy could be due to the fact that we assessed the levels of individual SCFAs at only one time point and dietary fiber consumption over a relatively short time. It is likely that a longer duration of high-fiber diet is needed for the changes in fecal SCFA levels to be observed. A healthy, balanced diet is an important component of IBD treatment as it allows patients to maintain a healthy body weight, which contributes to improved gut microbiota composition [31,32]. However, we did not show a correlation between the body mass index and the SCFA profile, while some studies indicated that underweight and obesity were involved in alterations of SCFA levels [33].

IBD is a heterogeneous group of gastrointestinal diseases, characterized by a different clinical course of the disease and therapeutic responses. Therefore, we investigated the relationship between the SCFA profile and various clinical features of patients, such as disease duration, IBD complications, and comorbidities, but no significant differences beyond those discussed were found. By using the PCA model, we achieved a reduction in dimensionality, that is, the original 12 parameters were replaced only by the first two principal components. As a result, coordinates of original parameters in the new coordinate system (loadings of the principal components) described the correlation structure in the data set. In addition, the use of the PCA model resulted in the conversion of a data matrix to a single informative plot, where the scatter of individual cases was easily visible. In this projection, some clusters of cases were revealed but they did not reflect different diagnoses.

The main limitations of our study are the small number of participants in study groups, a relatively large difference between the mean age of the study groups, and different treatment regimens in IBD patients. In addition, the nutritional analysis was conducted based on the frequency of consumption rather than a 24 h interview, which might have resulted in an underestimation of fiber intake. It is possible that due to all these limitations, there were no significant differences in the concentrations of the tested organic acids between active and inactive UC and CD, and in the control group. Additionally, further studies with a higher number of enrolled patients to support our preliminary observations are necessary. We are currently conducting further research in this area on a larger group of patients. We believe that by increasing the size of the study groups, we will obtain more accurate results that can be translated into clinical practice.

## 5. Conclusions

Our study confirmed that the described methodology for stool sample preparation and identification of acids is effective and can be used to determine the fecal levels of selected organic acids. Our findings demonstrated that medications used in IBD treatment may affect the fecal levels of SCFAs and BCFAs. Trimebutine use may increase propionic acid levels, while TNF-inhibitor therapy may lead to elevated isobutyric acid levels. Moreover, our study indicated a link between lactic acid levels and IBD severity. The possible underlying mechanism is that dysbiosis observed in active IBD may prevent conversion of lactate into butyrate acid. Further studies on a larger and more homogeneous group of participants are necessary to evaluate alterations in the levels of SCFAs and their precursors as well as associations between these alterations and any environmental factors in patients with CD and UC.

## Figures and Tables

**Figure 1 jcm-10-04701-f001:**
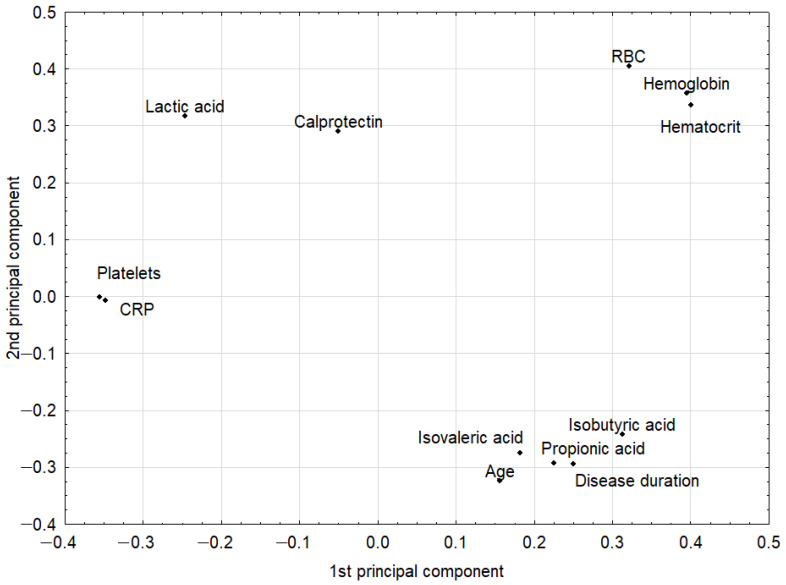
The loadings (coordinates of original parameters, i.e., the weights that combine original parameters with principal components) for the first two principal components in the principal component analysis (PCA) model. Original parameters with an absolute value of their weights higher than 0.3 were considered to be correlated with one another. CRP: C-reactive protein; RBC: red blood cells.

**Figure 2 jcm-10-04701-f002:**
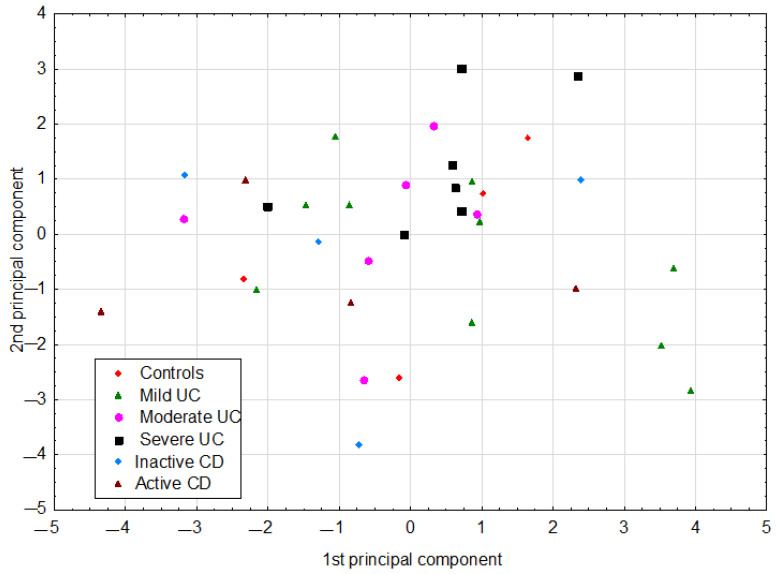
The scatter of individual cases (patients) on the plane defined by the first two principal components in the PCA model. CD: Crohn’s disease; UC: ulcerative colitis.

**Table 1 jcm-10-04701-t001:** Medical therapy and clinical characteristics of patients with ulcerative colitis (UC), Crohn’s disease (CD), and controls.

**Parameter**	**UC**	**CD**	**Controls**
No. of patients	23	8	4
Sex (male/female)	13/10 (57/43)	5/3 (63/37)	2/2 (50/50)
Median age, years	32	33.5	48
CDAI score (<150/≥150 points)	NA	4/4 (50/50)	NA
Mayo score (0/1/2/3)	0/10/6/7 (0/44/26/30)	NA	NA
**Therapy**			
Steroids (yes/no)	13/9 (59/41) ^1^	2/6 (25/75)	0/4 (0/100)
Azathioprine/6-mercaptopurine (yes/no)	9/14 (39/61)	5/3 (63/37)	0/4 (0/100)
5-Aminosalicylic acid (yes/no)	21/2 (91/9)	6/1 (86/14) ^1^	0/4 (0/100)
Biological drugs (yes/no)	2/21 (9/91)	3/5 (37/63)	0/4 (0/100)
Trimebutine (yes/no)	7/16 (30/70)	1/7 (12/88)	0/4 (0/100)

All results expressed in number (%) of patients. CDAI: Crohn’s Disease Activity Index; NA: not applicable. ^1^ Data missing for one patient.

**Table 2 jcm-10-04701-t002:** Concentrations [ug/g] of organic acids depending on disease severity in patients with UC, CD, and controls.

Organic Acid	Study Group						
	Controls (*n* = 4)	Mild UC (*n* = 10)	Moderate UC (*n* = 6)	Severe UC (*n* = 7)	Inactive CD (*n* = 4)	Active CD (*n* = 4)	*p* Value
Succinic [ug/g]	144.6 (77.0; 238.6)	472.4 (197.3; 1368.5)	193.3 (118.4; 279.1)	149.4 (125.3; 551.5)	195.3 (104.5; 456.8)	305.8 (147.1; 516.5)	0.495
Acetic [ug/g]	604.1 (446.7; 763.0)	709.0 (450.7; 967.4)	904.0 (552.4; 1082.3)	666.9 (594.0; 906.3)	648.0 (462.6; 742.6)	831.8 (523.3; 1202.0)	0.842
Lactic [ug/g]	128.3 (49.7; 843.9)	606.6 (216.5; 2286.9)	621.7 (119.8; 1591.0)	871.7 (294.3; 2616.2)	392.5 (26.5; 2077.1)	994.5 (334.1; 1858.6)	0.516
Propionic [ug/g]	181.5 (61.0; 331.8)	211.3 (117.8; 576.1)	365.5 (241.9; 548.1)	223.9 (71.1; 440.8)	218.1 (166.7; 457.6)	451.7 (205.9; 598.2)	0.593
Butyric [ug/g]	109.0 (67.5; 176.3)	226.2 (72.5; 375.4)	378.0 (144.4; 642.2)	157.5 (54.7; 375.4)	217.0 (109.4; 228.7)	147.2 (<LOD; 305.9)	0.768
Isobutyric [ug/g]	19.0 (<LOD; 50.1)	26.8 (<LOD; 61.8)	<LOD (<LOD; 49.9)	<LOD (<LOD; 42.4)	40.1 (21.4; 61.8)	21.9 (<LOD; 39.1)	0.902
Valeric [ug/g]	13.4 (<LOD; 31.1)	<LOD (<LOD; <LOD)	<LOD (<LOD; <LOD)	<LOD (<LOD; 29.7)	<LOD (<LOD; 20.3)	<LOD (<LOD; 24.9)	0.916
Isovaleric [ug/g]	41.4 (<LOD; 122.8)	29.7 (<LOD; 280.2)	24.5 (<LOD; 51.3)	<LOD (<LOD; 49.9)	42.1 (18.8; 67.0)	<LOD (<LOD; 20.3)	0.742

All results expressed in medians (lower and upper quartiles). Abbreviations: LOD, limit of detection.

**Table 3 jcm-10-04701-t003:** Correlation weights (expressing the strength of bivariate associations) for the pairs of original parameters used in the PCA model; only the strongest correlation weights with an absolute value equal or higher than 0.100 were shown.

Pairs of Correlated Parameters		Correlation Weights
Hemoglobin	Hematocrit	0.158
RBC	Hemoglobin	0.142
RBC	Hematocrit	0.133
RBC	Lactic acid	0.128
Platelets	CRP	0.124
Hemoglobin	Lactic acid	0.111
Hematocrit	Lactic acid	0.104
Hemoglobin	CRP	−0.100
Hemoglobin	Platelets	−0.104
Hematocrit	CRP	−0.105
Hematocrit	Platelets	−0.109

CRP: C—reactive protein; RBC: red blood cells.

## Data Availability

The data presented in this study are available on request from the corresponding author.

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
