# Peer review of "Fecal Levels of Lactic, Succinic and Short-Chain Fatty Acids in Patients with Ulcerative Colitis and Crohn Disease: A Pilot Study"

_jcm, 2021, doi:10.3390/jcm10204701_

Round 1
Reviewer 1 Report
In their manuscript entitled "Fecal Levels of Lactic, Succinic and Short-Chain Fatty Acids, in Patients with Ulcerative Colitis and Crohn Disease: A Pilot Study", Kaczmarczyk et al. present an observational study from one single clinical center. The authors evaluate the concentration of 8 specific organic acids in the stool of IBD patients and combined this with clinical parameters at the timepoint of collection.
Overall, the topic of the study is interesting and the presentation of the data adequate. The manuscript is concisely written and easy to understand.
However, the study and the manuscript has several limitations:
- The number of study participants is extremely low leading to a huge variation within the data. The authors show results which do not support other studies in the field. This is important and interesting, but I am in doubt that this point can be made on such a weak basis!
- The figures including PCAs do not visualize novel information and can be taken out.
Author Response
Comments to authors:
The number of study participants is extremely low leading to a huge variation within the data. The authors show results which do not support other studies in the field. This is important and interesting, but I am in doubt that this point can be made on such a weak basis!
Response:
Thank you for this comment. It is partially the limitation of our study. The low number of participants is due to the fact that the results we presented in this pilot study, are the beginning of our research. We are currently conducting research on a larger group of participants. Nevertheless, thanks to the pilot study, we were able to make interesting observations that require further precise investigation. However, despite the small number of enrolled patients some of our observations coincide with previous reports, which confirms the correctness of the research carried out. Our results, which are not in line with other researchers, are presented in the discussion and emphasized the preliminary nature of the reports. We have made modifications to the manuscript to highlight the initial nature of this observations.
Comments to authors:
The figures including PCAs do not visualize novel information and can be taken out.
Response:
Thank you for this suggestion. By using the PCA model, we achieved a reduction of dimensionality, that is, the original 12 parameters were replaced only by the first two principal components. As a result, coordinates of original parameters in the new coordinate system (loadings of the principal components) described the correlation structure in the data set. In addition, the use of the PCA model resulted in the conversion of a data matrix to a single informative plot, where the scatter of individual cases was easily visible. In this projection, some clusters of cases were revealed but they did not reflect different diagnoses. Hence it seems important to us to present them to the Readers.

Reviewer 2 Report
Olga Kaczmarczyk et al., study is very well performed. It is relevant to give us a better understanding of the IBD disease relapses and remission, dysbiosis, and diet. However, I have few corrections.
Line 35-38
Is this information worldwide?
Material method section 2.2 needs more detail on the filtration process.
Author Response
Comments to authors:
Line 35-38. Is this information worldwide?
Response:
Thank you for the noticeable inaccuracies in lines 35-38. We have made an appropriate correction to emphasize the global nature of the given IBD epidemiology.
Comments to authors:
Material method section 2.2 needs more detail on the filtration process.
Response:
Thank you for this comment. In part 2.2 “Measurement of Organic Acids in Stool Samples” we have added more details about filtration process to improve the quality of the manuscript.

Reviewer 3 Report
In this study, Olga Kaczmarczyk, et al. evaluated a profile of organic acids in the stool of Crohn’s Disease and Ulcerative Colitis patients with different disease activity. They showed higher propionic acid levels in IBD patients who received trimebutine and higher isobutyric acid levels in patients treated with biological drugs.
Strengths of this study:
Study question is valid.
Adequate literature review was performed.
The author's major findings were clearly presented. They adequately address the stated research objectives.
The research results validate the author's conclusions.
I have a few suggestions:
- Consider including P-values in Table 2.
- Authors noticed lower butyric acid concentrations in active CD and Severe UC. How will they explain this finding?
- Authors may want to include more details about clinical profile of IBD patients, like duration of disease, surgical details, other comorbidities like irritable bowel syndrome etc. which can alter organic acid profile.
Author Response
Comments to authors:
Consider including P-values in Table 2.
Response:
Thank you for your suggestion. We included p-values in Table 2 to improve the presentation of our results.
Comments to authors:
Authors noticed lower butyric acid concentrations in active CD and Severe UC. How will they explain this finding?
Response:
Thank you for this comment.
This phenomenon can be explained by the fact that the deficiency of certain beneficial microbes such as Faecalibacterium prausnitzii in IBD patients, especially those with severe course of disease, leads to a reduction in butyrate levels and contributes to the development and maintenance of inflammation in IBD. Butyric acid has a beneficial effect on the intestines, e.g. has anti-inflammatory and antioxidant properties, strengthens the intestinal barrier and nourishes the colonocytes. Therefore, its lower level may contribute to the exacerbation of the disease.
We made modifications to the manuscript in the discussion section to discuss this result.
Comments to authors:
Authors may want to include more details about clinical profile of IBD patients, like duration of disease, surgical details, other comorbidities like irritable bowel syndrome etc. which can alter organic acid profile.
Response:
Thank you for your suggestion. It is well-known that IBD is a heterogeneous disease in terms of disease course. Therefore, we investigated the relationship between the SCFA profile and various clinical features of patients, such as disease duration and comorbidities, but no significant differences beyond those discussed in manuscript were found. In addition, Figure 1 shows that we analyzed some of these factors. Also, we have added the appropriate paragraph in the discussion relating to this issue.

Round 2
Reviewer 3 Report
Authors addressed all my concerns.
Author Response
Thank you for this comment. We appreciate any suggestion and we are glad that the corrections met your expectations.